# Socio-ecological correlates of physical activity in breast and colon cancer survivors 4 years after participation in a randomized controlled exercise trial (PACT study)

Anouk E. Hiensch[1], Petra H. M. Peeters[1], Marijke Jansen[2,3], Elsken van der Wall[4], Frank J. G. Backx[5], Miranda J. Velthuis[6], Anne M. May[1] *

1 Julius Center for Health Sciences and Primary Care, University Medical Center Utrecht, Utrecht University, Utrecht, The Netherlands, 2 Department of Human Geography and Spatial Planning, Faculty of Geosciences, Utrecht University, Utrecht, The Netherlands, 3 Institute for Nursing Studies, Hogeschool Utrecht, Utrecht, The Netherlands, 4 Department of Medical Oncology, University Medical Center Utrecht, Utrecht University, Utrecht, The Netherlands, 5 Department of Rehabilitation, Physical Therapy Science & Sport, University Medical Center Utrecht, Utrecht University, Utrecht, The Netherlands, 6 Netherlands Comprehensive Cancer Organisation (IKNL), Utrecht, The Netherlands

* A.M.May@umcutrecht.nl

**Data Availability Statement:** All relevant data are within the manuscript and its Supporting Information files.

## Abstract

### Background

Having a physically active lifestyle after cancer diagnosis is beneficial for health, and this needs to be continued into survivorship to optimize long-term benefits. We found that patients, who participated in an 18-week exercise intervention, reported significant higher physical activity (PA) levels 4 years after participation in a randomized controlled trial of supervised exercise delivered during chemotherapy (PACT study). This study aimed to identify social-ecological correlates of PA levels in breast and colon cancer survivors 4 years after participation in the PACT study.

### Methods

Self-reported PA levels and potential correlates (e.g. physical fitness, fatigue, exercise history, and built environment) were assessed in 127 breast and colon cancer survivors shortly after diagnosis (baseline), post-intervention and 4 years later. Multivariable linear regression analyses were performed to identify social-ecological correlates of PA 4 years post-baseline.

### Results

The final model revealed that lower baseline physical fatigue (β = -0.25, 95% CI -0.26; -0.24) and higher baseline total PA (0.06, 95% CI, 0.03; 0.10) were correlated with higher total PA levels 4 years post-baseline. Higher baseline leisure and sport PA (0.02, 95% CI 0.01; 0.03), more recreational facilities within a buffer of 1 km (4.05, 95% CI = 1.28; 6.83), lower physical fatigue at 4-year follow-up (-8.07, 95% CI -14.00; -2.13), and having a

**Funding:** This work was supported by The Netherlands Organisation for Health Research and Development (ZonMw, project number: 171002202, https://www.zonmw.nl/nl/onderzoek-resultaten/doelmatigheidsonderzoek/programmas/project-detail/doelmatigheidsonderzoek/physical-activity-during-cancer-treatment-pact-studie-een-gerandomiseerde-studie-naar-de-effecten/, PHMP), the Dutch Cancer Society (KWF Kankerbestrijding. Project number: UU 2009-4473, PHMP), the Dutch Pink Ribbon Foundation (2011. WO02.C100, https://www.pinkribbon.nl/doelbestedingen/wetenschappelijk-onderzoek/2011/pact-studie.html, AMM) and VIOZ (Stichting Vrienden Integrale Oncologische Zorg (2015, MJV). The funders had no role in study design, data collection and analysis, decision to publish, or preparation of the manuscript.

**Competing interests:** The authors have declared that no competing interests exist.

positive change in physical fatigue during the intervention period (0.04, 95% CI 0.001; 0.07) were correlates of sport and leisure PA levels 4 years post-baseline.

## Conclusions

This study suggests that baseline and 4-year post-baseline physical fatigue, and past exercise behaviour, were significant correlates of PA 4 years after participation in an exercise trial. Additionally, this study suggests that the built environment should be taken into account when promoting PA. Understanding of socio-ecological correlates of PA can provide insights into how future exercise interventions should be designed to promote long-term exercise behaviour.

## Trial registration

Current Controlled Trials ISRCTN43801571, Dutch Trial Register NTR2138. Trial registered on 9 December 2009,

http://www.trialregister.nl/trialreg/admin/rctview.asp?TC=2138

## Background

Increasing evidence supports the role of physical activity (PA), both during and after cancer treatment, as an effective intervention to enhance health-related quality of life (HRQoL) [1], improve cardiorespiratory fitness [2], decrease cancer-related fatigue [3], and possibly even to improve survival and reduce the risk of recurrence [4,5]. Despite compelling evidence that PA is beneficial for cancer survivors, the proportion of breast and colon cancer survivors that do meet the exercise guidelines of performing at least 150 minutes per week of moderate-to-vigorous physical activity, varies from 15% to 44% [6–8]. Even up to 5–10 years post-diagnosis, both breast and colon cancer survivors have not fully returned to their pre-diagnosis level of PA [9,10].

Since sufficient evidence exists regarding the beneficial effects of exercise [7], effective interventions to maintain sufficient levels of PA during cancer survivorship should be pursued. Understanding of characteristics that influence long-term PA levels may facilitate the development of more effective and targeted exercise interventions or advices designed to promote long-term behaviour change.

Characteristics that influence exercise behaviour have been of interest to many researchers. Theoretical frameworks, such as the Theory of Planned Behavior (TPB) and Social Cognitive Theory, have been employed to examine correlates of exercise behaviour. These studies reported that younger age [11,12], lower body mass index [11], higher self-efficacy [11], positive attitude towards PA [12], and more social support [11] were significantly associated with higher PA levels in breast cancer survivors, whereas younger age [13], sufficient baseline PA levels [14], planning and intention as part of the TPB framework [13], and higher education [13] were significantly associated with higher PA levels in colon cancer survivors. While these correlates do have merit in explaining exercise behaviour, increasing evidence has shown that exercise behaviour might also be influenced by environmental factors. Proximity to potential destinations and the presence of bicycle lanes and parks have been most consistently associated with PA participation in the general population [15,16].

Hence, it would be more informative to adopt a broader theoretical framework, such as the social-ecological framework, to identify correlates of PA in this population [17]. Specifically, the social-ecological framework emphasizes the environmental context of exercise behaviour,

while incorporating psychosocial, demographical and physical influences to identify multiple correlates of PA. So far, studies have largely focused on the individual aspects of the social-ecological model with little emphasis on the objective built environment. Besides, little research has been performed on correlates of PA levels years after completion of cancer treatment. Mutrie *et al.* (2012) found that those who maintained a physically active lifestyle 5 years after cancer treatment still benefit in terms of higher levels of HRQoL and lower levels of depression [18]. Since cancer treatment is known to have long-lasting side effects, this physically active lifestyle needs to be maintained in order to optimize long-term benefits.

In the 'Physical Activity during Cancer Treatment' (PACT) study, a randomized controlled exercise trial, we demonstrated that an 18-week supervised exercise intervention (i.e. moderate-to-high intensity aerobic and resistance training), delivered during adjuvant cancer treatment, had significant beneficial short-term effects on physical fatigue, submaximal cardiorespiratory fitness and muscle strength in breast and colon cancer patients [19,20]. Furthermore, we found that patients, who participated in the PACT exercise intervention, reported significant higher physical activity levels four years after participation in the PACT study compared to the usual care group [21]. The purpose of the present study was to explore socio-demographic, clinical, psychosocial, physical and environmental correlates of subjectively assessed PA in breast and colon cancer survivors on average 4 years after participation in the PACT study.

## Methods

### Setting and participants

A detailed description of the PACT study design has been published previously [22]. In short, the original study was conducted in seven hospitals in the Netherlands between 2010 and 2013. This multicenter controlled trial randomly assigned 204 breast and 33 colon cancer patients to either usual care while maintaining their current PA pattern (*n* = 118) or to supervised aerobic and muscle strength training in addition to usual care (*n* = 119). Inclusion criteria were: a histological diagnosis of cancer less than six (breast cancer) or ten (colon cancer) weeks before study recruitment; stage M0; scheduled for chemotherapy; age 25–75 years; not treated for any cancer in the preceding 5 years (except basal skin cancer); able to read and understand the Dutch language; Karnovsky Performance Status of ≥60; able to walk 100 meter or more; and no contraindications for PA (as assessed through the Revised Physical Activity Readiness Questionnaire). After written informed consent was obtained, a concealed computer-generated randomization, following a 1:1 ratio and stratified for age (25–40, 40–65 and 65–75 years), adjuvant treatment (radiotherapy yes/no before chemotherapy), use of tissue expander (for patients with breast cancer), tumor type and hospital, was used to allocate patients to the two groups. The study was approved by the Medical Ethics Committee of the University Medical Centre Utrecht and the local Ethical Boards of the participating hospitals (07-271/O).

Three to four years after inclusion in the original PACT study, the treating physician approached 197 PACT participants again for information on their current health status. The present study was originally not planned when participants were recruited for the former PACT study and consent was asked again. Eleven participants were deceased or otherwise considered not healthy enough to participate by the treating oncologist (Fig 1). We did not invite participants who indicated not to be willing to fill in questionnaires at subsequent time points (*n* = 29). Participants who dropped out during the original PACT study, but indicated to be willing to fill in questionnaires at all subsequent time points were invited. Participants, who signed written informed consent, were asked to fill in questionnaires at home.

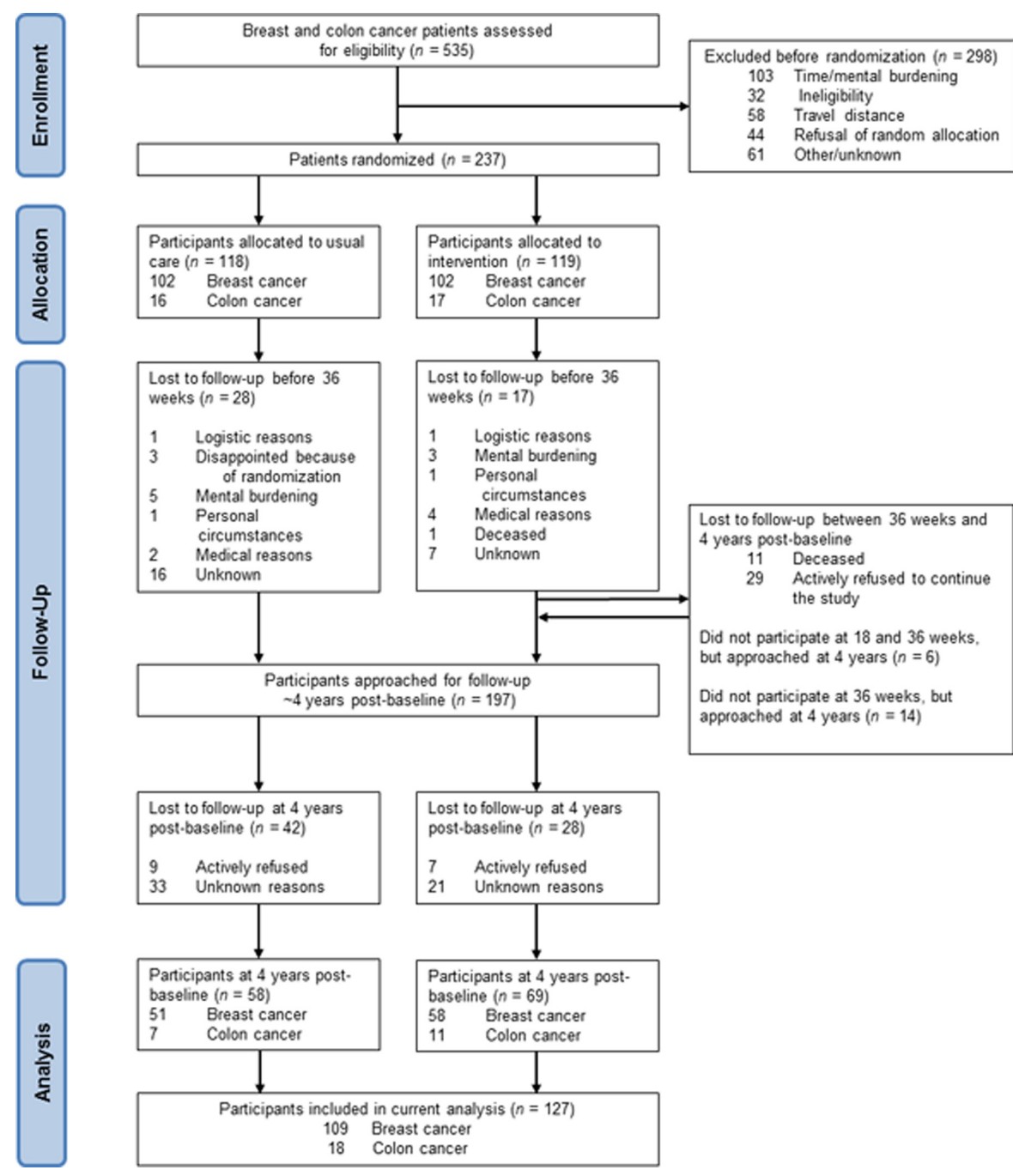

**Fig 1. Flow of participants through the PACT study.**

### Intervention (2010–2013)

The exercise intervention consisted of an 18-week supervised aerobic and muscle strength exercise program in addition to the usual care. Participants attended two 1-hour sessions per week, supervised by a physiotherapist. The aerobic and muscle strength exercises were individualized to the participants' preferences and fitness levels as assessed by means of a cardiopulmonary exercise test and one repetition maximum muscle strength tests. In addition to the

intervention, participants were asked to be physically active for at least 30 minutes a day, on three other days of the week, according to the Dutch Physical Activity Guidelines 2011 [23,24]. Principles of Bandura's social cognitive theory were incorporated to promote maintenance of a physically active lifestyle [25]. This theory emphasizes the role of cognitive processes in determining health behavior such as exercise. The most important construct of this theory is self-efficacy, which beliefs are based on actual/mastery experience, vicarious/observational learning, verbal persuasion and emotional arousal. In the PACT supervised exercise program, we addressed the first three determinants [22].

Participants in the control group received usual care and were asked to maintain their habitual PA pattern up to week 18. Thereafter, they were allowed, for ethical reasons, to participate in exercise programs offered in the Netherlands to cancer patients after completion of primary treatment.

## Outcome measure

**Physical activity.**    PA levels 4-years after participation in the PACT study were assessed using the Short Questionnaire to Assess Health-enhancing physical activity (SQUASH) [26]. This validated 4-item self-report instrument contains questions about commuting, leisure time and sports, and household activities and activities at work and school. For each activity, duration, frequency and intensity is assessed. Minutes per week of moderate-to-high intensity total PA and leisure and sport activity were calculated. Moderate-to-high intensity PA was defined as $\geq$ 4 metabolic equivalent (MET) [27].

## Candidate correlates

In order to explore correlates of PA 4 years post-diagnosis in breast and colon cancer survivors, data of the intervention and control group were combined. The intervention was added as a correlate to the model. Candidate correlates were selected based on literature and clinical reasoning and divided into four categories according to the social-ecological framework (Fig 2): (1) socio-demographic and clinical variables, (2) psychosocial, (3) physical, and (4) environmental factors.

**Socio-demographic and clinical factors.**    Socio-demographic factors were collected using a self-reporting questionnaire and included age at baseline (in years), marital status (married or single), education (low (i.e., elementary and lower vocational education), intermediate (i.e., secondary (vocational) education), and high (i.e., higher vocational and university education)), and employment status at 4-year follow-up. Clinical data were retrieved from medical records and included body mass index (BMI), tumor site, previous treatment with chemotherapy or in combination with radiotherapy and allocation to the exercise group or usual care.

**Psychosocial factors.**    Patient reported outcome measures included fatigue, HRQoL, and anxiety and depression. Fatigue was measured using the validated Dutch version of the Multi-dimensional Fatigue Inventory (MFI) [28]. The MFI is a 20-item questionnaire and consists of five dimensions: general fatigue, physical fatigue, reduced activity, reduced motivation and mental fatigue. The minimum and maximum scores are respectively 4 and 20, with higher scores indicating more fatigue.

The HRQoL was assessed using the validated 30-item European Organization for Research and Treatment of Cancer Quality of Life Questionnaire (EORTC-QOL-C30) [29]. The EORTC-QOL-C30 incorporates five functional scales (physical, role, emotional, cognitive and social functioning), one quality of life scale and one symptom scale (including pain and fatigue). All scales ranging from 0 to 100. A high scale score represents a higher response level.

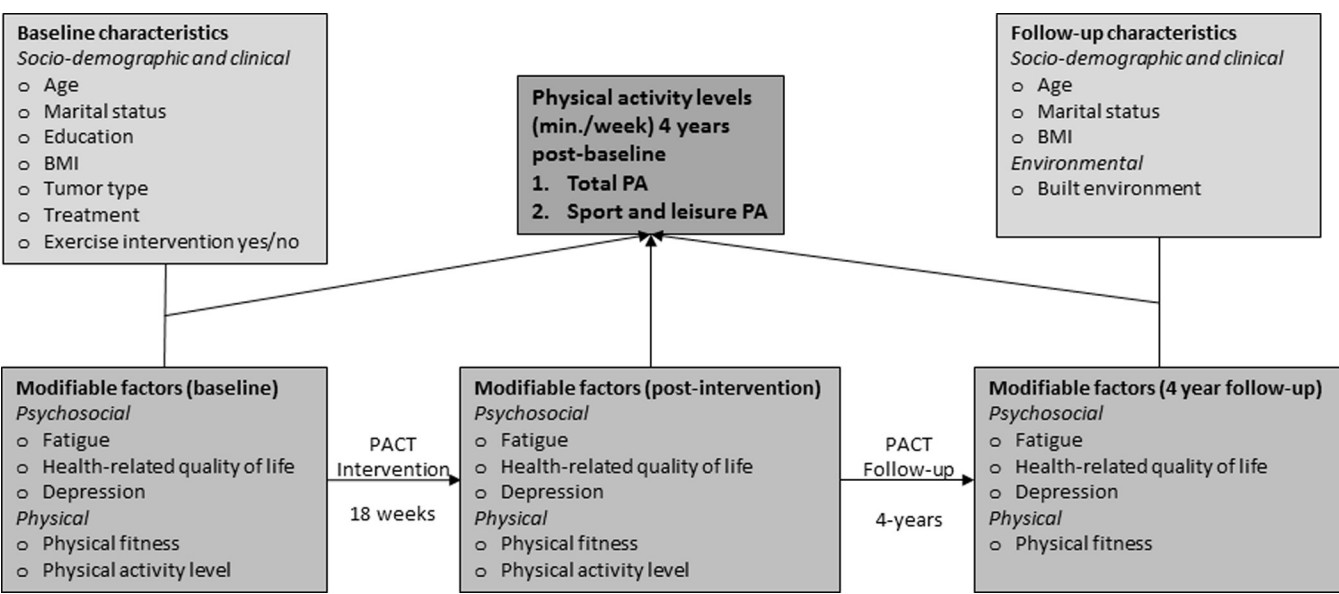

**Fig 2. Conceptual model of physical activity levels in breast and colon cancer survivors 4 years post-diagnosis.**

Anxiety and depression were assessed using the validated Dutch version of the Hospital Anxiety and Depression Scale (HADS), with scores ranging from 0–21 and higher scores indicating more anxiety and depression [30].

**Physical fitness.** Correlates related to physical fitness included baseline PA, aerobic capacity, muscle strength of the quadriceps and handgrip strength. Baseline PA was assessed in the same way as the outcome by using the SQUASH and we asked participants to fill in their PA level for a usual week in the months preceding study entry. Aerobic capacity was determined during a cardiopulmonary exercise test with continuous breathing gas analysis. Peak oxygen uptake ($VO_{2peak}$) was calculated by taking the mean of $VO_2$-values measured in the last 30 seconds before exhaustion. Muscle strength of the quadriceps was assessed using a Cybex dynamometer at an angular velocity of 60˚/s for both legs. The highest peak torque of three repetitions was calculated. Handgrip strength of both hands was recorded in kilogram force (kgF) by a mechanical handgrip dynamometer. Handgrip strength was obtained by taking the best score of two attempts.

**Built environment.** Postal codes of participants' home were obtained through a questionnaire. Coordinates of these postal codes were uploaded in ArcGIS, and Euclidean buffers of 1 and 5 km were drawn around each coordinate. Objective physical environmental characteristics of these buffers were obtained from the 2012 Land Use data from the Statistics Netherlands and Accommodation Monitor Sport (AMS) data from the Mulier Institute, using ArcGIS software. The percentage of land area comprised of both total green and open spaces for recreation, and residential area, and the number of private recreation facilities were determined within both buffers.

Green and open spaces included public parks, forest, playgrounds, nature trails, green belts, beaches and dunes, whereas residential area included primary housing facilities. Private recreation facilities, including accommodations where PA could occur and required payment (e.g. fitness centres, swimming pools and dance studios), were identified and geocoded using data from the Mulier Institute. Accommodations for physical education purposes only, motorsports, indoor skiing, martial arts and extreme sports were excluded from this database. For

each participant, the number of private recreation facilities within a 1- and 5-km buffer of their home was calculated.

## Measurement points

The outcome and candidate correlates were assessed at baseline (shortly after diagnosis), 18 weeks (post-intervention) and 36 weeks (post-baseline) in the original PACT study. In addition, follow-up measurements of the outcome and candidate correlates, except for correlates related to physical fitness, were performed at a median of 4 year post-baseline. In the present study, only correlates assessed at baseline, post-intervention and 4 years post-baseline were taken into account. Change scores were computed for all continuous variables by subtracting the baseline score from the post-intervention score. These scores were also considered candidate correlates.

## Missing data

Relatively little data was missing on the outcome PA (2.4% on both total and leisure and sport PA) and candidate correlates (4.7%); no data were missing on correlates of built environment. Nonetheless, to address potential selection bias, missing values were imputed by chained equations using R package MICE [31]. In this case, the Missing at Random assumption was found to be plausible. In total, 20 complete datasets were produced and subsequently analyses were conducted in each of the imputation sets. Estimates from each of these sets were combined using Rubin's rules to account for variation within and between datasets.

## Statistical analysis

Baseline demographics were summarised for all breast and colon cancer patients together and compared for inactive, intermediate active and active participants [27]. The minutes per week spent on both moderate-to-high intensity total PA and moderate-to-high intensity leisure and sport activity, measured 4 year post-baseline, were considered the main outcomes. Since the distribution of these continuous variables was highly skewed, with the presence of many zeroes and extreme scores, the normality assumption was violated. Therefore, the outcomes were transformed using the Box-Cox method [32].

Univariable associations between candidate correlates and total and leisure and sport PA were estimated with linear regression analyses. Relevant correlates of PA levels were selected when $p < 0.20$ [33]. Prior to multivariable regression analyses, multicollinearity between the selected correlates was checked. The multivariable linear regression analyses included a backward stepwise selection procedure to identify correlates for the final multivariable model. Selection was based on a $p$ value of 0.05. Correlates for the final multivariable model were selected using the majority method, which means that included correlates were selected in at least 10 of 20 imputation sets [34]. Based on our previous finding that patients, who participated in an 18-week exercise intervention, reported significant higher PA levels 4 years after participation in the PACT study [21], randomization was a priori added to the multivariable model. The regression coefficients of the multivariable model were estimated in each imputation set separately and combined using Rubin's rules in order to produce the regression coefficients of the final multivariable model. Accordingly, the regression coefficients (β) with corresponding 95% confidence interval and the explained variance ($R^2$) of the final models were reported. All statistical analyses were performed in R version 3.3.2.

## Results

### Participants

Fig 1 shows the CONSORT flow diagram of the PACT study. Between January 2010 and December 2012, a total of 237 patients with breast and colon cancer were included in the original PACT study. Four years post-baseline, 197 PACT participants were eligible and approached to participate in the 4 year post-baseline measurements, and 127 (64%) participants signed informed consent. Sixteen PACT participants refused to participate in the 4-year post-baseline measurements and 54 PACT participants did not respond for unknown reasons, also after one reminder had been sent. Baseline characteristics were not significantly different between participating patients and non-participating eligible patients (p > 0.05) (21).

Characteristics of all PACT breast and colon cancer survivors, who participated in the 4 year post-baseline measurements, are presented in Table 1 according to three PA levels (i.e., inactive, semi-active and active based on the Dutch Physical Activity Guidelines 2011(≥150 min per week of moderate-to-vigorous physical activity) [27]). Being inactive was defined as not engaging in any physical activity, whereas semi-active and active were defined as being active on a moderate-to-vigorous intensity on 1–4 days per week and on at least 5 days per week, respectively [27]. The 127 study participants were predominantly diagnosed with breast cancer (85.8%), married (81.1%), employed (64.6%) and had a mean age of 55.8 (SD = 8.2) years. The median number of minutes per week spent on moderate-to-vigorous intensity total PA was 420 (range 160.0 to 810.0), whereas the median number of minutes per week spent on moderate-to-vigorous sport and leisure PA was 194 (range 120.0 to 380.0). Based on the Dutch Physical Activity Guidelines 2011, 6 (4.7%) participants were categorized as being inactive, 47 (37.0%) as being intermediate active and 74 (58.3%) as being active.

### Correlates of physical activity 4 years post-baseline

The results of the univariable and multivariable linear regression analyses are presented in Additional file 1 and Table 2, respectively. Fig 3 provides a graphical overview of the multivariable models to facilitate interpretation.

Two correlates were identified to be associated with total PA: lower baseline physical fatigue (β = -0.25, 95% CI -0.26; -0.24) and higher baseline total PA (0.06, 95% CI, 0.03; 0.10). Collectively, the correlate randomization together with these two correlates explained 31.2% of the variance in total PA.

The multivariable model revealed that experiencing less physical fatigue at 4 year follow-up (-8.07, 95% CI -14.00; -2.13), having a positive change in physical fatigue during the intervention period (0.04, 95% CI 0.001; 0.07), having a higher number of private recreational facilities within a buffer of 1 km (4.05, 95% CI 1.28; 6.83), and being physically active at baseline (0.02, 95% CI 0.01; 0.03) were correlates of sport and leisure PA at 4 year follow-up. Collectively, the correlate randomization together with these four variables explained 29.0% of the variance in the outcome variable.

## Discussion

The present study explored psychosocial, physical and environmental correlates of moderate-to-vigorous total and sport and leisure PA levels 4 years after participation in a randomized controlled trial of supervised exercise delivered during chemotherapy in breast and colon cancer patients. Our results indicate that higher levels of moderate-to-vigorous total PA 4 years post-baseline were more likely among breast and colon cancer survivors who were less fatigued shortly after diagnosis (baseline) and had higher baseline total PA levels. Yet, higher levels of

**Table 1. Characteristics of PACT breast and colon cancer survivors according to PA levels ~4 years post-diagnosis.**

| | All participants (n = 127) | Inactive participants[a] (n = 6) | Semi-active participants[b] (n = 47) | Active participants[c] (n = 74) |
|---|---|---|---|---|
| **Socio-demographical characteristics** | | | | |
| Age (years) | 55.8 ± 8.2 | 56.0 ± 6.0 | 56.1 ± 8.9 | 55.5 ± 7.9 |
| Sex (female %) | 117 (92.1) | 6 (100.0) | 39 (83.0) | 72 (97.3) |
| Marital status | | | | |
| Married | 103 (81.1) | 5 (83.3) | 41 (87.2) | 57 (77.0) |
| Single | 24 (18.9) | 1 (16.7) | 6 (12.8) | 17 (23.0) |
| Education | | | | |
| Low | 10 (7.9) | 0 (0.0) | 6 (12.8) | 4 (5.4) |
| Medium | 58 (45.7) | 3 (50.0) | 18 (38.3) | 37 (50.0) |
| High | 59 (46.5) | 3 (50.0) | 23 (48.9) | 33 (44.6) |
| BMI (kg/m$^2$) | 26 ± 4.1 | 25.6 ± 6.1 | 25.7 ± 3.7 | 26.2 ± 4.1 |
| Employed | 82 (64.6) | 5 (83.3) | 32 (68.1) | 45 (60.8) |
| **Clinical characteristics** | | | | |
| Tumor site | | | | |
| Breast cancer | 109 (85.8) | 6 (100.0) | 38 (80.9) | 65 (87.8) |
| Colon cancer | 18 (14.2) | 0 (0.0) | 9 (19.1) | 9 (12.2) |
| Radiotherapy | 79 (62.2) | 2 (33.3) | 28 (59.6) | 49 (66.2) |
| Tumor receptor status | | | | |
| Triple negative | 19 (15.0) | 1 (16.7) | 7 (14.9) | 11 (14.9) |
| Her2+, ER or PR | 15 (11.8) | 0 (0.0) | 3 (6.4) | 12 (16.2) |
| Her2+, ER & PR- | 9 (7.1) | 0 (0.0) | 5 (10.6) | 4 (5.4) |
| Her2-, ER or PR+ | 66 (52.0) | 5 (83.3) | 23 (48.9) | 38 (51.4) |
| **Psychosocial characteristics** | | | | |
| *Multidimensional Fatigue Inventory* | | | | |
| General fatigue | 9.8 ± 4.1 | 10.2 ± 5.9 | 10.3 ± 4.2 | 9.4 ± 4.0 |
| Physical fatigue | 9.4 ± 4.2 | 10.2 ± 5.7 | 9.7 ± 4.2 | 9.1 ± 4.2 |
| Reduced activity | 8.5 ± 4.0 | 9.5 ± 6.4 | 8.9 ± 3.9 | 8.1 ± 3.8 |
| *EORTC QLQ C-30* | | | | |
| Global health status | 77.4 ± 16.9 | 70.8 ± 28.3 | 76.8 ± 16.1 | 78.4 ± 16.4 |
| Physical functioning | 88.6 ± 12.7 | 76.0 ± 28.9 | 88.9 ± 10.2 | 89.2 ± 12.3 |
| Social functioning | 89.4 ± 18.0 | 77.8 ± 40.4 | 89.4 ± 16.5 | 90.4 ± 16.2 |
| Emotional functioning | 83.3 ± 19.3 | 73.6 ± 37.8 | 82.8 ± 18.8 | 84.5 ± 17.6 |
| Pain | 15.4 ± 21.8 | 30.6 ± 37.1 | 12.4 ± 21.0 | 16.0 ± 20.5 |
| **Physical characteristics** | | | | |
| *SQUASH (median, IQR)* | | | | |
| Moderate-to-vigorous intensity total PA (min/week) | 420.0 (160.0–810.0) | 20.0 (0.0–50.0) | 240.0 (130.0–300.0) | 615.0 (420.0–1222.5) |
| Moderate-to-vigorous intensity leisure and sport PA (min/week)[d] | 195.0 (120.0–380.0) | 10.0 (0.0–27.5) | 120.0 (50.0–180.0) | 360.0 (186.8–480.0) |
| **Environmental characteristics** | | | | |
| Number of private recreation facilities <1km | 9.5 ± 7.5 | 9.0 ± 7.1 | 9.6 ± 7.2 | 9.5 ± 7.8 |
| *Land use[e] (%)* | | | | |
| Residential area | 37.3 ± 19.2 | 47.0 ± 15.4 | 32.4 ± 15.5 | 39.6 ± 21.0 |
| Total green and open space (median, IQR) | 5.3 (2.4–10.2) | 7.3 (4.0–13.6) | 5.3 (2.5–9.3) | 5.7 (2.2–10.2) |
| **Group allocation** | | | | |
| Usual care | 58 (45.7) | 4 (66.7) | 21 (44.7) | 33 (44.6) |

*(Continued)*

**Table 1.** (Continued)

| | All participants (*n* = 127) | Inactive participants[a] (*n* = 6) | Semi-active participants[b] (*n* = 47) | Active participants[c] (*n* = 74) |
|---|---|---|---|---|
| Exercise intervention | 69 (54.3) | 2 (33.3) | 26 (55.3) | 41 (55.4) |

*Abbreviations*: BMI Body Mass Index, *PA* physical activity, *IQR* interquartile range.

Continuous variables are presented as mean ± SD, whereas dichotomous or categorical variables are presented as *n* (%) unless stated otherwise.

[a]Inactive is defined as '0 days/week of 30 min activity',

[b]semi-active is defined as '1–4 days/week of 30 min activity', and

[c]active is defined as '≥5 days/week of 30 min activity' (32).

[d]Including work, leisure and sport activities. [e]Land use indicates the percentage socioeconomic use of land within a buffer (i.e. < 1km)

moderate-to-vigorous leisure and sport PA were more likely among breast and colon cancer survivors who were less fatigued at 4 year follow-up, had a positive change in physical fatigue during the intervention period, had a higher number of private recreation facilities in the neighbourhood and higher baseline sport and leisure PA levels.

Furthermore, this study showed that breast and colon cancer survivors reported spending 195 (120–380) min/week on moderate-to-vigorous sport and leisure PA and 420 (160–810) min/week on moderate-to-vigorous total PA 4 years after participation in the PACT study. A recent meta-analysis of 19 interventions targeting physical activity in cancer survivors reported that the majority of included studies observed lower levels of moderate-to-vigorous total PA [35]. At 6-month to 5-year follow-up, the minutes per week spent on moderate-to-vigorous total PA varied from 37 to 556 min/week. It should be noted that the majority of studies, contrary toi our study, excluded patients based on current activity levels (e.g. <150 min/week) and that different types of interventions were investigated in terms of setting, mode of delivery, and length of the intervention. As a consequence, direct comparison of PA levels might be hampered. =

**Table 2.** Correlates of moderate-to-vigorous total and sport and leisure PA levels 4 years post-baseline.

| | Correlates at baseline, during the intervention, and 4 year follow-up |
|---|---|
| | β (95% CI) |
| **Correlates of Total PA** | |
| Randomization to usual care | -1.03 (-5.89; 3.82) |
| MFI–Baseline physical fatigue | -0.25 (-0.26; -0.24) |
| Total PA at baseline[a] | 0.06 (0.03; 0.10) |
| $R^2$ | 31.2 |
| **Correlates of Sport and Leisure PA** | |
| Randomization to usual care | -0.99 (-5.84; 3.87) |
| MFI–Physical fatigue at 4 year follow-up[b] | -8.07 (-14.00;-2.13) |
| MFI–Change in physical fatigue during the intervention period | 0.04 (0.001;0.07) |
| Number of private recreation facilities <1km[b] | 4.05 (1.28;6.83) |
| Sport and Leisure PA at baseline | 0.02 (0.01;0.03) |
| $R^2$ | 29.0 |

*Abbreviations*: PA physical activity, MFI multifactorial fatigue inventory

[a]Baseline total PA was transformed using the box-cox method, whereas

[b]physical fatigue at 4 year follow-up and the number of private recreation facilities <1 km were log transformed.

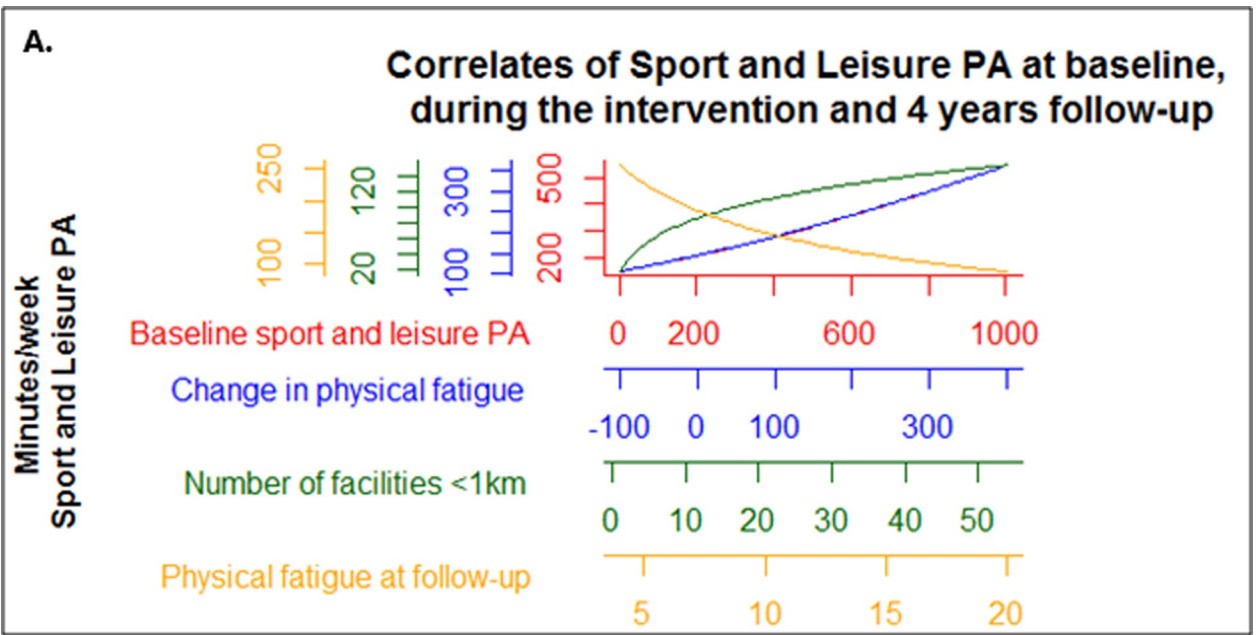

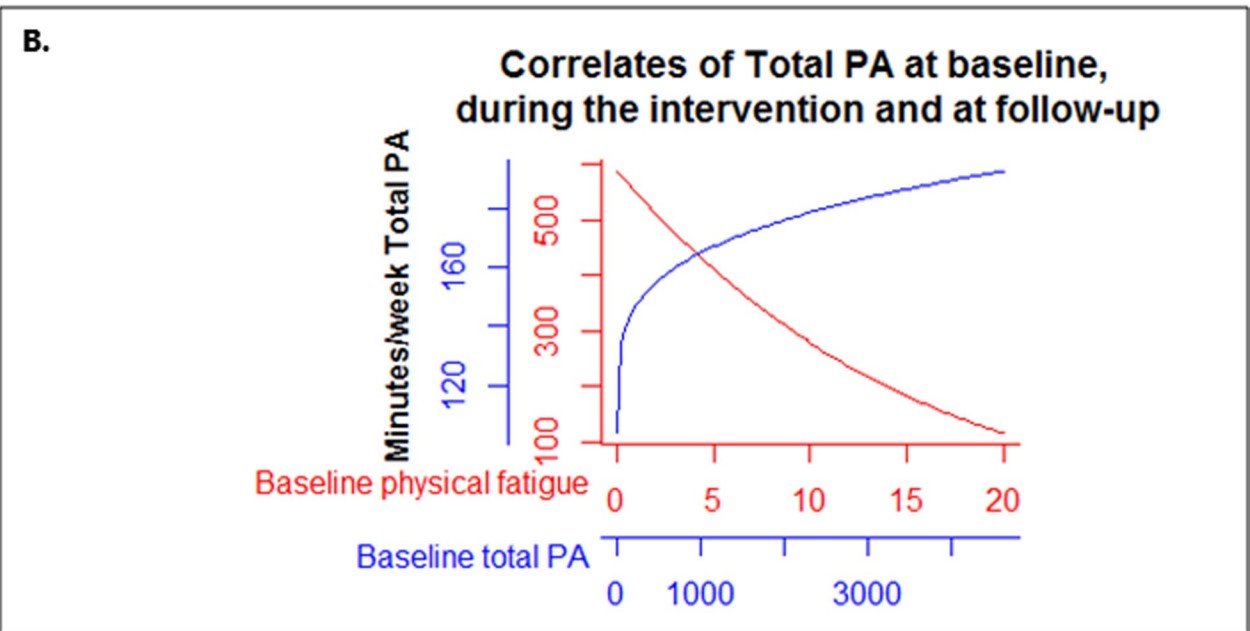

**Fig 3. Correlates of both total and sport and leisure PA 4 years post-diagnosis.** Graphical representation of the association between correlates and (A.) moderate-to-vigorous sport and leisure and (B.) total physical activity levels (min/week) to simplify interpretation. All correlates have their own x- and y-axes. Note that the x-axes also contain extreme values.

Consistent with earlier findings in cancer survivors [12, 14,36,37], the strongest correlate of PA at 4 year follow-up in our study was baseline PA. It is not surprising that breast and colon cancer survivors who participated in physical activities prior to chemotherapy would be more likely to return to their habitual PA level after chemotherapy, since pre- and post-trial exercise are more likely to be self-motivated, whereas exercise during the intervention period was supervised. In our recent publication, we showed that patients who participated in the PACT

exercise intervention, reported significant higher physical activity levels four years after partic-ipation in the PACT study [21]. Please note that in the current univariable analyses, randomi-zation was not a significant correlate of PA. This can be attributed to the fact that we only included participants who participated in the 4 year post-baseline measurements instead of all PACT participants in our previous publication. Furthermore, in our primary analysis, inten-tion-to-treat mixed linear regression models were used to model PA over time, adjusted for the baseline value of the outcome and other prognostic factors [21].

Another finding of our study was that baseline physical fatigue was a significant correlate of total PA levels at 4 year follow-up and that baseline and 4 year post-baseline physical fatigue was a correlate of sport and leisure PA levels. Comparable with these findings, Courneya et al. (2009) reported that post-intervention fatigue was associated with PA levels at 6 months follow-up in a longitudinal study, whereas Lee et al. (2016) observed that baseline fatigue was associated with PA maintenance 6 months post-intervention [12,38]. Furthermore, we found that cancer survi-vors who experience a positive change in physical fatigue during the intervention period are more likely to be physically active at 4 year follow-up. This result demonstrates the importance of exercising during chemotherapy. Given that maintenance of sufficient PA is important for the effectiveness of an exercise intervention, it would be essential for clinicians and physiotherapists to provide targeted support to those with high baseline fatigue based on these findings.

Finally, this study suggests that the built environment should be taken into account when promoting physical activity. Specifically, the number of private recreation facilities in the neighbourhood was a significant correlate of sport and leisure PA levels. However, it should be noted that the association between the built environment and sport and leisure PA may be confounded by direct (i.e. preference for neighbourhoods with recreational facilities) and indi-rect self-selection measures (socio-economic status). Our finding parallels results from previ-ous studies in the general population where the presence of destinations linked to PA (e.g., indoor and outdoor recreational facilities) was associated with higher PA levels [16]. Other studies in healthy adults have found different environmental correlates of PA levels, including the spatial accessibility of shops and recreational facilities [16], safe bicycle and walking routes [39], greenness [15] and walkability [15]. In contrast to James et al. (2017), this study did not observe an association between total green and open space and PA, which might be due to the use of a different definition of green space exposure.

A few comparable studies in cancer survivors examined the correlates of meeting PA guide-lines or PA levels. Trinh et al. (2016) found that meeting PA guidelines was associated with the perceived presence of many retail shops in the neighbourhood in kidney cancer survivors [40]. Lynch et al. (2010) observed that the built environment was associated with achieving suffi-cient levels of PA 5 months post-diagnosis [41], whereas Kampshoff et al. (2016) showed that breast cancer survivors from urban areas were more likely to be physically active compared to breast cancer survivors from rural areas [11]. These findings support the theory that environ-ments which facilitate PA may motivate people to be more physically active.

The remaining correlates that were considered in the present study showed no significant association with PA levels, including age, marital and employment status, level of education and treatment-related characteristics. Previous studies examining socio-demographical and clinical characteristics and physical activity report mixed results [11,12, 40,42]. The fact that these socio-demographic and clinical variables are not associated with PA levels suggest that, based on the correlates included in this study, no additional subgroups at risk for low PA levels can be identified. In addition, one might argue that the incorporation of the principles of Ban-dura's social cognitive theory in the PACT exercise intervention might have helped partici-pants to maintain a physically active lifestyle after study participation. As a result, physical

activity might have become a habit independent of socio-demographical or clinical characteristics.

The final multivariable models explained 31.2% of the variance in moderate-to-vigorous total PA and 29.0% of the variance in moderate-to-vigorous sport and leisure PA, indicating that part of the variance in the outcome could be explained by the social-ecological correlates in the models. Nonetheless, it also indicates that there might be other variables explaining PA levels, for instance financial situation and the presence of safe bicycle lanes and walking routes, that were not included in the present study, but are important to explain differences in PA levels among breast and colon cancer survivors.

Our results should be viewed within the context of important strengths and limitations. This is the first study to prospectively examine the correlates of PA 4 years after participation in an exercise trial in breast and colon cancer survivors. Most of the work in this field relies on cross-sectional studies in which causal relationships cannot be implied. Future studies should consider a longitudinal study design to get more insight into the social-ecological correlates of PA levels, and to verify the findings of this study. Another strong feature of this study is the assessment of a broad range of correlates by adopting a theoretical social-ecological framework. Limitations of our study include the reliance on self-reported measures of PA, since it is prone to over-reporting. Objective measurement of PA would overcome this limitation and would provide a more valid estimate of PA in future studies. Nonetheless, subjectively measured PA using the SQUASH has been shown to be reliable [26]. Given the large number of comparisons made in this exploratory study and the relatively small sample size, this may have increased the chance of finding spurious correlates. Nevertheless, many of these findings are consistent with previous research examining correlates of PA among patients with cancer. Finally, the transformation of the outcome could have introduced complexity in the substantive interpretation of the final model as this changed the nature of the variables. Due to the non-linear relationship between the correlates and PA, we provided a graphical overview of the multivariable models to facilitate interpretation.

## Conclusions

The present study contributes to new insights on how social-ecological correlates are related to PA levels 4 years after participation in an exercise trial in breast and colon cancer survivors. In this study population, PA levels were explained by a range of variables including psychosocial, physical and environmental variables. These characteristics should be considered when designing future exercise interventions or advices to promote long-term exercise behaviour in breast and colon cancer survivors, and subsequently, to optimize long-term benefits of PA.

## Supporting information

**S1 Table. Univariable associations between candidate social-ecological correlates and moderate-to-vigorous total physical activity levels (min/week) and moderate-to-vigorous sport and leisure PA (min/week).**
(DOC)

**S1 File. Onderzoeksprotocol Op weg naar herstel_amendement 261110.**
(PDF)

**S2 File. CONSORT checklist.**
(DOC)

## Acknowledgments

We would like to thank the participants and the professional staff at St. Antonius Hospital, Nieuwegein and Utrecht; Diakonessen Hospital, Utrecht; Meander Medical Centre, Amersfoort; Rivierenland Hospital, Tiel; Orbis Medical Centre, Sittard; Zuwe Hofpoort Hospital, Woerden and University Medical Center Utrecht, The Netherlands. Their participation made this study possible.

## Author Contributions

**Conceptualization:** Petra H. M. Peeters, Miranda J. Velthuis, Anne M. May.

**Data curation:** Anne M. May.

**Formal analysis:** Anouk E. Hiensch, Marijke Jansen.

**Funding acquisition:** Petra H. M. Peeters, Miranda J. Velthuis, Anne M. May.

**Investigation:** Marijke Jansen, Elsken van der Wall, Frank J. G. Backx, Miranda J. Velthuis, Anne M. May.

**Methodology:** Anouk E. Hiensch.

**Project administration:** Petra H. M. Peeters, Anne M. May.

**Supervision:** Petra H. M. Peeters, Anne M. May.

**Writing – original draft:** Anouk E. Hiensch.

**Writing – review & editing:** Petra H. M. Peeters, Marijke Jansen, Elsken van der Wall, Frank J. G. Backx, Miranda J. Velthuis, Anne M. May.

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
