## [Decision Letter · Decision Letter 0]

5 Feb 2020

PONE-D-19-35637

Socio-ecological correlates of physical activity in breast and colon cancer survivors 4 years after participation in a randomized controlled exercise trial (PACT study)

PLOS ONE

Dear MSc Hiensch,

Thank you for submitting your manuscript to PLOS ONE. After careful consideration, we feel that it has merit but does not fully meet PLOS ONE’s publication criteria as it currently stands. Therefore, we invite you to submit a revised version of the manuscript that addresses the points raised during the review process.

We would appreciate receiving your revised manuscript by Mar 21 2020 11:59PM. To enhance the reproducibility of your results, we recommend that if applicable you deposit your laboratory protocols in protocols.io, where a protocol can be assigned its own identifier (DOI) such that it can be cited independently in the future. For instructions see: http://journals.plos.org/plosone/s/submission-guidelines#loc-laboratory-protocols

We look forward to receiving your revised manuscript.

Kind regards,

Justin C. Brown

Academic Editor

PLOS ONE

Reviewers' comments:

Reviewer's Responses to Questions

**Comments to the Author**

1. Is the manuscript technically sound, and do the data support the conclusions?

Reviewer #1: Yes

Reviewer #2: Yes

2. Has the statistical analysis been performed appropriately and rigorously? 

Reviewer #1: Yes

Reviewer #2: Yes

3. Have the authors made all data underlying the findings in their manuscript fully available?

Reviewer #1: Yes

Reviewer #2: Yes

4. Is the manuscript presented in an intelligible fashion and written in standard English?

Reviewer #1: Yes

Reviewer #2: Yes

5. Review Comments to the Author

Reviewer #1: This study is a secondary analysis of long-term outcomes of the PACT study. The original study found those participating in the PACT study had significantly higher PA levels after 4 years. The aim to explore the personal and environmental correlates of this finding. Follow up surveys were administered to eligible participants. Strongest correlate found was baseline PA. This study adds to the literature describing correlates of exercise among people with cancer. The novel aspect is the long-term analyses. However, some inclusion or mention of habit formation and automaticity would help the discussion since the paper is about long-term behaviour change.

Overall, great paper. Some specific comments to follow.

Major comments

Page 18 line 338. Why was ≥4 METs used rather than ≥3 METs?

Minor Comments

Page 7 line 147. The sentence describing the frequency of sessions is a bit awkward.

Page 7 line 152. Which principles of SCT were incorporated? I realise this is the original study description, but could be briefly described.

Page 20 line 380. Could comment on the different between built environment and physical environment more broadly. Rural areas with trails and green space better than those without? Or is it just man-made facilities that make the difference?

Page 20 line 385. Long-term PA behaviour not associated with age, marital and employment status, education, treatment-related variables is an interesting result. Potential discussion point for habit formation and automaticity?

Reviewer #2: PONE-D-19-35637

February 3, 2020

The authors report on an interesting study examining correlates of physical activity in cancer survivors four years after participation in an RCT for an exercise intervention. There are just a few minor points that need addressing.

Abstract – Suggest changing the first word of the conclusions. Intuitive doesn’t really make sense.

Page 11, Statistical Analyses section – Did the authors control for variables like age, education, and BMI in the multivariable models?

Page 11, lines 255-256 – It would be helpful to have a reference here to indicate where the authors came up with the justification to use the cut point of p < .20.

Page 13, lines 286-288 – The authors should explain what the Dutch healthy exercise norms are. If they are the same 150 mins/week as stated in the Introduction it should be indicated there that this is the Dutch guidelines, or described in the Results section.

Page 18, lines 332-340 – the authors compare the study participants’ PA levels to other studies of colorectal and breast cancer survivors, but are those other studies not cohort studies of free living cancer survivors in the population? They are not comparable to a study of highly motivated participants self-selected to be part of and then underwent an 18 week intervention. It would be more effective to compare these findings to other follow up studies of interventions for cancer survivors.

Page 20, lines 366-373 – it would be fair to make a statement in this paragraph that perhaps what we are really observing is difference in socioeconomic status? At least in North America, high SES neighbourhoods have more recreational facilities available, and people of high SES are healthier and more active.

Table 1: Is there not income status to report or some measure of socioeconomic status?

Table 2: Formatting is required.

6. PLOS authors have the option to publish the peer review history of their article (what does this mean?). If published, this will include your full peer review and any attached files.

Reviewer #1: No

Reviewer #2: No

---

## [Author Response · Author response to Decision Letter 0]

27 Mar 2020

Response to the reviewers: PONE-D-19-35637

“Socio-ecological correlates of physical activity in breast and colon cancer survivors 4 years after participation in a randomized controlled exercise trial (PACT study)”

We would like to thank the editor for the opportunity to provide a response to the thoughtful comments and suggestions raised by the reviewers. We include a point-by-point response to their comments and the changes in the manuscript are highlighted with ‘track changes’. We hope that these revisions will make the manuscript suitable for publication in your journal.

Report reviewer #1:

Reviewer #1: This study is a secondary analysis of long-term outcomes of the PACT study. The original study found those participating in the PACT study had significantly higher PA levels after 4 years. The aim to explore the personal and environmental correlates of this finding. Follow up surveys were administered to eligible participants. Strongest correlate found was baseline PA. This study adds to the literature describing correlates of exercise among people with cancer. The novel aspect is the long-term analyses. However, some inclusion or mention of habit formation and automaticity would help the discussion since the paper is about long-term behaviour change.

Overall, great paper. Some specific comments to follow.

Major comments

1. Page 18 line 338. Why was ≥4 METs used rather than ≥3 METs?

Authors’ answer to comment 1:

We defined moderate-to-high intensity activities according to the Dutch Physical Activity Guidelines, including only activities with a MET-value of at least 4.0 (Hildebrandt et al., 2011). We have now added this reference to the manuscript as well. Note that these guidelines have been updated in the meanwhile (i.e. after the 4-year follow-up measurements) (Weggemans et al., 2018). Since the PACT study was conducted before this update, we still refer to the Dutch Physical Activity Guidelines of 2011.

Minor Comments

2. Page 7 line 147. The sentence describing the frequency of sessions is a bit awkward.

Authors’ answer to comment 2: 

We agree with the reviewer that this sentence doesn’t read well. We have now reformulated this sentence: ‘Participants attended two 1-hour sessions per week, supervised by a physiotherapist.’

3. Page 7 line 152. Which principles of SCT were incorporated? I realise this is the original study description, but could be briefly described.

Authors’ answer to comment 3:

We have now elaborated a bit further on the principles of SCT (page 7, line 153-157): ‘Principles of Bandura’s social cognitive theory were incorporated to promote maintenance of a physically active lifestyle. This theory emphasizes the role of cognitive processes in determining health behavior such as exercise. The most important construct of this theory is self-efficacy, which beliefs are based on actual/mastery experience, vicarious/observational learning, verbal persuasion and emotional arousal. In the PACT supervised exercise program, we addressed the first three determinants (Velthuis et al., 2010).’

4. Page 20 line 380. Could comment on the different between built environment and physical environment more broadly. Rural areas with trails and green space better than those without? Or is it just man-made facilities that make the difference?

Authors’ answer to comment 4:

There is indeed a difference between built and physical environment. The built environment is part of the physical environment, which is constructed by human activity, whereas the physical environment includes both the natural and built environment. The built environment for example includes homes, schools, cycle lanes, parks and recreational areas. In our manuscript we used both terms. The present study mainly focusses on the built environment (i.e. recreational facilities, parks and residential area), so we refer to the built environment when we discuss findings of the current study. However, in our discussion we mention other studies who use the term physical environment, while they actually investigate the built environment. We have now used the term ‘built environment’ throughout the whole manuscript. 

We did not assess differences between the green space and built environment, therefore we cannot draw conclusions. Based on our study, we might conclude that man-made facilities play an important role in stimulating physical activity. We found an association between the number of recreational space and sport and leisure PA, but not for green space. We have now added this finding to the paragraph (page 21, line 400-402). In previous studies, green space has been inconsistently associated with physical activity, because studies use different definitions of green space exposure. 

5. Page 20 line 385. Long-term PA behaviour not associated with age, marital and employment status, education, treatment-related variables is an interesting result. Potential discussion point for habit formation and automaticity?

Authors’ answer to comment 5:

We would like to thank the reviewer for this interesting discussion point. Long-term physical activity behavior requires physical activity to be systematically imbedded in one’s daily life and to be part of someone’s behavior. The PACT exercise intervention incorporated principles of Bandura’s social cognitive theory to help participants maintain a physically active lifestyle after cancer treatment. This might have resulted in physical activity to become a habit independent of socio-demographical or medical factors. We have added this discussion point and the paragraph now reads as follows: 

‘The remaining correlates that were considered in the present study showed no significant association with PA levels, including age, marital and employment status, level of education and treatment-related characteristics. Previous studies examining socio-demographical and clinical characteristics and physical activity report mixed results. The fact that these socio-demographic and clinical variables are not associated with PA levels suggest that, based on the correlates included in this study, no additional subgroups at risk for low PA levels can be identified. In addition, one might argue that the incorporation of the principles of Bandura’s social cognitive theory in the PACT exercise intervention might have helped participants to maintain a physically active lifestyle after study participation. As a result, physical activity might have become a habit independent of socio-demographical or clinical characteristics.’

Report Reviewer #2: PONE-D-19-35637

February 3, 2020

The authors report on an interesting study examining correlates of physical activity in cancer survivors four years after participation in an RCT for an exercise intervention. There are just a few minor points that need addressing.

1. Abstract – Suggest changing the first word of the conclusions. Intuitive doesn’t really make sense.

Authors’ answer to comment 1:

We have now changed the first word of the conclusions. 

2. Page 11, Statistical Analyses section – Did the authors control for variables like age, education, and BMI in the multivariable models?

Authors’ answer to comment 2:

We did not control for variables like age, education and BMI in the multivariable models, since we were interested in whether these variables were correlates of long-term physical activity levels. Therefore, we decided to add them as correlates to the model. Found determinants are independent of the other factors included in the model related to 4-year PA level. 

3. Page 11, lines 255-256 – It would be helpful to have a reference here to indicate where the authors came up with the justification to use the cut point of p < .20.

Authors’ answer to comment 3:

We have now added a reference to justify the cut-off point of p<.20. 

4. Page 13, lines 286-288 – The authors should explain what the Dutch healthy exercise norms are. If they are the same 150 mins/week as stated in the Introduction it should be indicated there that this is the Dutch guidelines, or described in the Results section.

Authors’ answer to comment 4:

We would like to thank the reviewer for this relevant suggestion. We agree that an explanation of the Dutch Physical Activity Guidelines would be helpful. In our introduction (page 4, line 74-75), we refer to both Dutch and non-Dutch studies, therefore, we chose to not explicitly state that they meet the Dutch Physical Activity Guidelines. Nevertheless, the advices are the same. We have now added an explanation of the Dutch Physical Activity Guidelines to the results section (page 13, line 285-289). The specific paragraph now reads as: ‘Characteristics of all PACT breast and colon cancer survivors, who participated in the 4 year post-baseline measurements, are presented in Table 1 according to three PA levels (i.e., inactive, semi-active and active based on the Dutch Physical Activity Guidelines 2011 (≥150 min per week of moderate-to-vigorous physical activity)). Being inactive was defined as not engaging in any physical activity, whereas semi-inactive and active were defined as being active on a moderate-to-vigorous intensity on 1-4 days per week and on at least 5 days per week, respectively.’

5. Page 18, lines 332-340 – the authors compare the study participants’ PA levels to other studies of colorectal and breast cancer survivors, but are those other studies not cohort studies of free living cancer survivors in the population? They are not comparable to a study of highly motivated participants self-selected to be part of and then underwent an 18 week intervention. It would be more effective to compare these findings to other follow up studies of interventions for cancer survivors.

Authors’ answer to comment 5:

These studies are indeed cohort studies of free living cancer survivors in the population. Therefore, we agree with the reviewer that these cancer survivors are not comparable to our study participants, including self-selected participants who are motivated to be allocated to an 18-week exercise intervention. We have now compared our findings to other follow-up studies of exercise interventions for cancer survivors using a recently published systematic review and meta-analysis by Grimmett et al. (2019). The paragraph now reads as (page 19, line 350-358): 

‘Furthermore, this study showed that breast and colon cancer survivors reported spending 195 (120-380) min/week on moderate-to-vigorous sport and leisure PA and 420 (160-810) min/week on moderate-to-vigorous total PA 4 years after participation in the PACT study. A recent meta-analysis of 19 interventions targeting physical activity in cancer survivors reported that the majority of included studies observed lower levels of moderate-to-vigorous total PA (Grimmett et al., 2019). At 6-month to 5-year follow-up, the minutes per week spent on moderate-to-vigorous total PA varied from 37 to 556 min/week. It should be noted that the majority of studies, contrary to our study, excluded patients based on current activity levels (e.g. <150 min/week) and that different types of interventions were investigated in terms of setting, mode of delivery, and length of the intervention. As a consequence, direct comparison of PA levels might be hampered.’ 

6. Page 20, lines 366-373 – it would be fair to make a statement in this paragraph that perhaps what we are really observing is difference in socioeconomic status? At least in North America, high SES neighbourhoods have more recreational facilities available, and people of high SES are healthier and more active.

Authors’ answer to comment 6:

We would like to thank the reviewer for this relevant suggestion. We agree that the association we find between the number of recreational activities and physical activity might be confounded by indirect self-selection (i.e. socio-economic status). Indeed, a cross-sectional study conducted in five European urban regions observed that perceived availability of recreational facilities was associated with leisure-time PA (Mackenbach et al., 2018). However, the association was attenuated after adjusting for direct self-selection measures (i.e. preference for neighborhoods with recreational facilities), whereas indirect self-selection (i.e. education as an indicator for socio-economic status) did not play a significant role. This suggests that we might be observing a difference in preferences, but not socio-economic status per se. Nevertheless, it should be noted that the crude measure of education might not be a good reflection of socio-economic constraints. We have now added the following sentence to this paragraph: ‘However, it should be noted that the association between the built environment and sport and leisure PA may be confounded by direct (i.e. preference for neighborhoods with recreational facilities) and indirect self-selection measures (socio-economic status).’

7. Table 1: Is there not income status to report or some measure of socioeconomic status?

Authors’ answer to comment 7:

Unfortunately, we are not able to report on income status, since no questions on income status were included in our questionnaires. Nevertheless, we reported on level of education and employment status, which can be regarded as a measure of socio-economic status. 

8. Table 2: Formatting is required.

Authors’ answer to comment 7:

The table has now been reformatted. 

References

Grimmett, C., Corbett, T., Brunet, J., Shepherd, J., Pinto, B. M., May, C. R., & Foster, C. (2019). Systematic review and meta-analysis of maintenance of physical activity behaviour change in cancer survivors. International Journal of Behavioral Nutrition and Physical Activity, 16(1), 37. https://doi.org/10.1186/s12966-019-0787-4

Hildebrandt, V. H., Bernaards, C. M., & Stubbe, J. H. (2011). Trendrapport Bewegen en Gezondheid 2010/2011.

Mackenbach, J. D., Matias de Pinho, M. G., Faber, E., Braver, N. den, de Groot, R., Charreire, H., Oppert, J.-M., Bardos, H., Rutter, H., Compernolle, S., De Bourdeaudhuij, I., & Lakerveld, J. (2018). Exploring the cross-sectional association between outdoor recreational facilities and leisure-time physical activity: the role of usage and residential self-selection. The International Journal of Behavioral Nutrition and Physical Activity, 15(1), 55. https://doi.org/10.1186/s12966-018-0689-x

Velthuis, M. J., May, A. M., Koppejan-Rensenbrink, R. A., Gijsen, B. C. M., van Breda, E., de Wit, G. A., Schröder, C. D., Monninkhof, E. M., Lindeman, E., van der Wall, E., & Peeters, P. H. M. (2010). Physical Activity during Cancer Treatment (PACT) Study: design of a randomised clinical trial. BMC Cancer, 10, 272. https://doi.org/10.1186/1471-2407-10-272

Weggemans, R. M., Backx, F. J. G., Chinapaw, M., Hopman, M. T. E., Koster, A., Kremers, S., Van Loon, L. J. C., May, A., Mosterd, A., van der Ploeg, H. P., Takken, T., Visser, M., Wendel-Vos, G. C. W., de Geus, E. J. C., & 2017, C. D. P. A. G. (2018). The 2017 Dutch Physical Activity Guidelines. International Journal of Behavioral Nutrition and Physical Activity, 15(58).

---

## [Editor Report · Decision Letter 1]

30 Mar 2020

Socio-ecological correlates of physical activity in breast and colon cancer survivors 4 years after participation in a randomized controlled exercise trial (PACT study)

PONE-D-19-35637R1

Dear Dr. Hiensch,

We are pleased to inform you that your manuscript has been judged scientifically suitable for publication and will be formally accepted for publication once it complies with all outstanding technical requirements.

With kind regards,

Justin C. Brown

Academic Editor

PLOS ONE
---

## [Editor Report · Acceptance letter]

2 Apr 2020

PONE-D-19-35637R1 

Socio-ecological correlates of physical activity in breast and colon cancer survivors 4 years after participation in a randomized controlled exercise trial (PACT study) 

Dear Dr. Hiensch:

I am pleased to inform you that your manuscript has been deemed suitable for publication in PLOS ONE. Congratulations! Your manuscript is now with our production department. 

With kind regards,

on behalf of

Dr. Justin C. Brown 

Academic Editor

PLOS ONE